# Investigation of Hydrothermal Performance in Micro-Channel Heat Sink with Periodic Rectangular Fins

**DOI:** 10.3390/mi14101818

**Published:** 2023-09-23

**Authors:** Heng Zhao, Honghua Ma, Xiang Yan, Huaqing Yu, Yongjun Xiao, Xiao Xiao, Hui Liu

**Affiliations:** 1School of Physics and Electronic Information Engineering, Hubei Engineering University, Xiaogan 432000, China; zhaoheng168188@163.com (H.Z.);; 2Wuhan National Laboratory for Optoelectronics (WNLO), Huazhong University of Science and Technology, Wuhan 430074, China; 3Institute of Engineering and Technology, Hubei University of Science and Technology, Xianning 437100, China

**Keywords:** micro-channel heat sink (MCHS), laminar flow, computational fluid dynamics

## Abstract

The micro-channel heat sink (MCHS) is an excellent choice due to its exceptional cooling capabilities, surpassing those of its competitors. In this research paper, a computational fluid dynamics analysis was performed to investigate the laminar flow and heat transfer characteristics of five different configurations of a variable geometry rectangular fin. The study utilized a water-cooled smooth MCHS as the basis. The results indicate that the micro-channel heat sink with a variable geometry rectangular fin has better heat dissipation capacity than a straight-type micro-channel heat sink, but at the same time, it has larger pressure loss. Based on the analysis of various rectangular fin shapes and Reynolds numbers in this study, the micro-channel heat sink with rectangular fins exhibits Nusselt numbers and friction factors that are 1.40–2.02 and 2.64–4.33 times higher, respectively, compared to the smooth heat sink. This significant improvement in performance results in performance evaluation criteria ranging from 1.23–1.95. Further, it is found that at a relatively small Reynolds number, the micro-channel heat sink with a variable geometry rectangular fin has obvious advantages in terms of overall cooling performance. Meanwhile, this advantage will decrease when the Reynolds number is relatively large.

## 1. Introduction

With the development of microfabrication technology, more and more electronic gadgets and microelectronics represent an irreversible change in high power, high heat dissipation, and miniaturization, especially in the fields of computing, automobile, telecommunication, and aerospace industries [1,2,3]. The significant challenge in the miniaturization of semiconductor products arises from the substantial heat generated within a limited space. Advanced electronic devices and microelectronics of the new generation are expected to produce heat dissipation in the range of multiple kilowatts, potentially reaching up to 1000 W/cm^2^ [4]. The operational temperature of microelectronic devices is influenced by their physical properties. For every 1 °C increase within the working temperature range of 70–80 °C, the reliability of these devices decreases by 5%. Additionally, a 10 °C rise in the junction temperature of electronic components leads to a 50% increase in the failure rate [5]. Many conventional heat removal technologies cannot effectively enhance the performance of heat transfer under the condition of heat flux of more than 100 W/cm^2^ [6]. Hence, the efficient thermal management of microelectronic devices is essential, considering overheating is harmful to the efficiency and reliability of microelectronic components. As a result, developing an efficient heat dissipation solution becomes a top priority. In recent years, the MCHS has emerged as the predominant heat dissipation method for semiconductor devices, particularly in the field of thermal solutions. The MCHS design consists of multiple parallel coolant micro-channels with varying widths, effectively reducing the thickness of the thermal boundary and significantly increasing the heat exchange surface area.

The single-layered MCHS was initially developed by Tuckerman in 1981. This heat sink design has the capability to dissipate heat at a rate of 790 W/cm^2^ under a temperature difference of 71 K between the inlet and outlet [7]. The primary purpose of the MCHS is to enhance the natural and forced convection’s ability to transfer heat. A study conducted by Adham focused on investigating the pressure drop and heat transfer characteristics of MCHS. The methodologies were also evaluated by the researchers. These approaches were employed to assess the overall performance of micro-channel heat sinks under various conditions of physical property parameters [8].

In light of the growing emphasis on size reduction and stringent temperature limitations in integrated micro-cooling systems, micro-channel heat sinks with passive microstructures are considered an efficient solution to meet these demands. This is because they do not require any external energy source, making them highly favorable in terms of energy efficiency. Xu [9,10] conducted a series of experiments and simulations to investigate the heat transfer characteristics of a micro-channel heat sink. This particular heat sink configuration consisted of parallel longitudinal micro-channels and multiple transverse microchambers. The findings revealed that the heat sink design was able to effectively reduce the pressure drop while enhancing heat transfer performance. This enhancement can be attributed to the reduced effective flow distance within the micro-channels. In their study, Cai et al. [11] investigated the impact of micro-channel geometry, rectangular ribs, and rib height on the overall performance of a heat sink. The researchers conducted a comparison between interrupted micro-channel heat sinks that incorporated rectangular ribs in transverse chambers and conventional heat sinks to analyze the obtained results. By evaluating the performance based on specific criteria, they aimed to determine whether the interrupted micro-channel heat sinks with rectangular ribs offered a superior cooling solution. Cheng [12] numerically simulated the effects of varying the microstructures on the thermal performance of the heat sink. They found that increasing the number of microstructures in each layer of the heat sink improved the thermal performance, resulting in a decrease in the overall temperature of the heat sink. In Xia’s study [13], the objective was to investigate the influence of structural parameters on the heat transfer rate and fluid flow within a system. The results revealed that changes in these parameters, such as the presence of reentrant cavities and the occurrence of jet and throttling effects, led to a notable improvement in the system’s performance. Specifically, the slipping of the working fluid over the reentrant cavities and the resulting jet and throttling effects played a crucial role in enhancing heat transfer and fluid flow within the system. This was achieved by allowing the fluid to flow more efficiently, creating a smoother and more efficient flow. Sui and Mohammed [14,15,16] conducted comprehensive studies involving both experimental and numerical approaches to investigate the laminar flow and heat transfer characteristics in wavy micro-channel heat sinks. Their research aimed to understand the behavior of fluid flow and heat transfer performance in these specific heat sink configurations. Their findings suggest that wavy micro-channel heat sinks have the potential to be more effective at dissipating heat than smooth micro-channel heat sinks. This is because the wavy channels create more turbulence, which increases the number of heat transfer points and increases the overall heat transfer rate. Additionally, the wavy channels also create more surface area, which increases the rate of heat transfer from the channel walls to the surrounding environment. Qu [17] analyzed the micro-pin–fin heat sink and the straight channel micro-channel heat sink and found that the straight channel micro-channel heat sink has a higher thermal resistance but a lower pressure loss. The two types of heat sinks are suitable for different applications. For applications where heat dissipation is the priority, the micro-pin–fin heat sink is a better choice. Rahbarshahlan et al. [18] improved heat transfer by adding hydrophobic surfaces to parts of the micro-channel, and research shows that the hydrophobic surfaces can reduce the amount of friction between the liquid and the surface, which further increases the efficiency of the heat transfer process. Zhang [19] enhanced heat transfer by the nanofluid-cooled heat sink. They obtained that reducing the diameter of the nanoparticles can increase the surface area, which can increase the number of contact points between the nanoparticles, resulting in better heat transfer performance. Increasing the volume fraction of the nanoparticles can also increase the number of contact points, which can improve the thermal conductivity of the heat sink.

Ghani et al. [20] explored the integration of sinusoidal cavities and rectangular ribs in the design of micro-channel heat sinks (MCHS). The results of their investigation demonstrated that the inclusion of sinusoidal cavities effectively increased the flow area, thereby minimizing pressure drop. Additionally, the presence of rectangular ribs enhanced the Nusselt number by reducing flow obstruction and promoting better heat transfer within the heat sink. Wang et al. [21] performed an experimental study to investigate the impact of microscale ribs and grooves on the performance of MCHS, with the Nusselt number enhanced up to 1.55 times that of a smooth channel. Zhai et al. [22] performed numerical investigations to analyze the performance of micro-channel heat sinks (MCHS) with various geometric structures of cavities and ribs. Their research aimed to understand how these variations in cavity and rib designs influenced the overall performance of the heat sink. The triangular cavities and ribs were found to be more effective than the other shapes. The triangular cavities and ribs provided higher thermal conductivity and better heat transfer characteristics compared to the other shapes. This is because of the increased surface area of the triangular cavities and ribs which allowed for more efficient heat transfer from the fluid to the walls of the micro-channel. Alfellag et al. [23] conducted numerical simulations to explore the fluid flow and heat transfer characteristics in a micro-channel heat sink featuring trapezoidal chambers and oval fins, both with and without slots. They found that the suggested design exhibited a pin aspect ratio of 1.25, a pin distance from the cavity center of 0.03 mm, and a slot thickness of 0.008 mm. Consequently, this design fulfilled a higher performance assessment requirement of 1.37. Based on the studies mentioned above, it is evident that the inclusion of ribs and cavities in micro-channel heat sinks improves heat transfer performance but also increases pumping power requirements. Extensive research has been conducted on conventional rib designs, but limited work has been found regarding the use of unique rib shapes.

In this investigation, a numerical simulation is performed to analyze the laminar flow and heat transfer within a micro-channel heat sink with variable geometry rectangular fins. This study presents the first reported performance analysis of such a structural arrangement, which has the potential to greatly improve thermal dissipation. The objective of this research is to compare the Nusselt number, performance evaluation criteria, fluid flow characteristics, pressure distribution, and temperature distribution with those of a conventional straight micro-channel heat sink (MCHS). The addition of a rectangular fin is expected to significantly improve MCHS’s overall performance.

## 2. Numerical Approach

### 2.1. Conservation Equations

Using a laminar flow, incompressible, steady-state, and three-dimensional model, the fluid flow within the micro-channel heat sink (MCHS) is simulated. The model was proposed by Zhang [24]. The model considers the effect of wall conduction on the velocity profile and the effect of fluid axial conduction on the temperature profile.

In order to simplify this numerical model, several assumptions are made as follows:The flow is the Newtonian incompressible laminar flow that is steady and continuous.Volume force, surface tension, and radiation heat transfer are not considered.Thermophysical properties are constant for the solid domain.

According to the assumptions made in this study, the numerical model incorporates the following equations for energy, continuity, and momentum, which are applicable to different micro-channel configurations [11].

Continuity equation:(1)$$\frac{\partial }{\partial \chi_{i}}(\rho_{f}u_{i})=0$$

Momentum equation:(2)$$\frac{\partial }{\partial \chi_{i}}(\rho_{f}u_{i}u_{j})=-\frac{\partial p}{\partial \chi_{j}}+\frac{\partial }{\partial \chi_{i}}[\mu_{f}(\frac{\partial u_{j}}{\partial \chi_{i}}+\frac{\partial u_{i}}{\partial \chi_{j}})]$$

Energy equation:(3)$$\frac{\partial }{\partial \chi_{i}}(\rho_{f}u_{i}c_{pf}T)=\frac{\partial }{\partial \chi_{i}}(k_{f}\frac{\partial T}{\partial \chi_{i}})+\mu_{f}[2{\left(\frac{\partial u_{i}}{\partial \chi_{i}}\right)}^{2}+{\left(\frac{\partial u_{j}}{\partial \chi_{i}}+\frac{\partial u_{i}}{\partial \chi_{j}}\right)}^{2}]$$

For the solid:(4)$$\frac{\partial }{\partial \chi_{i}}(k_{s}\frac{\partial T}{\partial \chi_{i}})=0$$

Here, *χ*_1_, *χ*_2_, and *χ*_3_ represent the x, y, and z coordinates, respectively. *ρ*, *μ* and *c_pf_* is density, dynamic viscosity, and specific heat capacity, respectively. *k* is thermal conductivity. Subscripts *f* and *s* refer to fluid and solid, respectively, as shown in Figure 1.

### 2.2. Boundary Conditions

In the system modeling, a uniform heat source with a power density of 400 KW/m^2^ is applied to the bottom surface of the substrate, specifically where the heat-generating chip is positioned. The top wall surface of the channels and the outer surfaces of the heat sink are considered to have thermal insulation. In contrast, the remaining walls that are in contact with the fluid are thermally linked through solid–fluid thermal conduction. A uniform velocity condition is assumed at the inlet, and a pressure condition is applied at the outlet (assumed as gauge pressure). To simplify the simulations, assume a uniform inlet temperature and apply no-slip boundary conditions to all walls [11].

This paper utilizes FLUENT 19.2, a software based on the finite volume method (FVM), to investigate the flow and heat transfer characteristics of the micro-channel heat sink (MCHS). To solve the coupled fluid–solid heat transfer problem, the same SIMPLEC algorithm employed in the previous literature is employed to address the heat transfer between the fluid and solid surfaces. The momentum and energy equations were discretized using a second-order upwind scheme. The chosen methodology is based on its ability to achieve rapid convergence to the numerical model. The solutions are deemed to have converged when the residuals for continuity, velocity, and energy equations are below 10^−6^, 10^−6^, and 10^−7^, respectively. Figure 1 illustrates the schematic diagram of the conventional rectangular micro-channel heat sink geometry. The parameters considered are within the following ranges: *T_in_* = 293 *K*, *P_out_* = 0 (gauge pressure), and *q_w_* = 400 KW/m^2^. Water and silicon are employed as the working fluid and solid material, respectively. The techniques for enhancing convective heat transfer can be categorized into passive techniques, active techniques, and combined enhancement methods [11]. Commonly used passive methods to enhance convective heat transfer include extended surfaces, flow disturbance devices, and flow channel structures [11]. The simulated model presented in Figure 2 of this paper is based on the aforementioned passive design principles. The structural design of cases 1 to 4 not only increases the surface area for heat transfer between the cold and hot fluids within the flow channel but also incorporates interrupted structures that induce bending of the fluid, promoting secondary flow and generating strong vortices even in laminar flow conditions. Figure 2 presents the detailed dimensions for case 0–case 4, while Table 1 provides the specific parameters for the external dimensions. Case 1 in Figure 2 involves adding a rectangular fin with a length of L_B_ inside the channel. Case 2 builds upon case 1 by incorporating periodic grooves into the rectangular fin with a length of L_B2_ and a depth of 1/2W_B_. Case 3 optimizes the rectangular fin by introducing a break with a length of L_B4_. Case 4 further optimizes case 3 by applying periodic breaks with a length of L_B4_. The design of these grooves and breaks in case 1–case 4 generates vortices in the opposite direction to the mainstream flow within the channel. These vortices effectively mix the hot and cold fluids, continuously disrupting the thermal boundary layer and enhancing heat transfer capability. The thermophysical properties of the working fluid are obtained from the corresponding reference [11].

### 2.3. Data Acquisition

In the current work, the MCHS is characterized by the following parameters, which govern fluid flow and heat transfer within the channels.

The Reynolds number is a dimensionless quantity that is defined as:(5)$$Re=\frac{\rho_{f}u_{m}D_{h}}{\mu_{f}}$$
where *ρ_f_* represents the volume average fluid density, *u_m_* is the average flow velocity in the smooth channel section, *D_h_* is the hydraulic diameter, and *μ_f_* denotes the dynamic viscosity of the fluid.
(6)$$D_{h}=\frac{2H_{c}W_{c}}{H_{c}+W_{c}}$$

The Reynolds number, at which laminar flow changes to turbulent flow is known as the critical Reynolds number, and previous studies have shown that laminar flow is common in micro-channels and that the critical Reynolds number is between 1000 and 1500 [25]. Therefore, in the numerical simulation of this paper, the laminar flow model was chosen for the numerical calculation, and the Reynolds number of fluid flow was kept below 1000.

The pressure drop (Δ*P*) is defined as the difference in pressure across the length of the micro-channel.
(7)$$\Delta p=\Delta p_{in}-\Delta p_{out}$$

Δ*p_in_* and Δ*p_out_* are the mass-weighted average inlet and outlet pressure. The average friction factor can be defined as:(8)$$\bar{f}=\frac{\Delta pD_{h}}{2\rho_{f}Lu_{m}^{2}}$$
where *L* denotes the whole length of the channel.

The average heat transfer coefficient is determined by the following formula:(9)$$h=\frac{q_{w}A_{b}}{2(W_{c}+H_{c})L_{}\Delta T}$$

*q_w_* is heat transfer through the base, *A_b_* is an area of the base, and Δ*T* is the temperature difference between the wall and fluid.
(10)$$\Delta T=T_{w}-T_{f}$$

Nusselt number *Nu* is a dimensionless number that represents the intensity of convective heat transfer and is also a standard for judging the performance of fluid heat transfer. Its expression is as follows:(11)$$Nu=\frac{hD_{h}}{k_{f}}$$

The *PEC* is given by:(12)$$PEC=\frac{Nu/Nu_{0}}{{\left(f/f_{0}\right)}^{1/3}}$$

Within this framework, the average heat transfer coefficient is determined by utilizing the Nusselt number (*Nu*_0_) and friction factor (*f*_0_) of the smooth micro-channel heat sink for calculation purposes.

### 2.4. Meshing and Grid Independent Test

To ensure the reliability of the simulations conducted in this study, the mesh independence assessment is performed. The rectangular straight micro-channel (referred to as case 0) is employed to assess the mesh independence. Different mesh sizes are employed and described in the provided table. Figure 3 illustrates the unstructured grid employed in this numerical simulation, which is refined locally using the meshing method.

As a result, four sets of unstructured grids generated by the pre-processing software ANSYS MESHING 2022 are evaluated, namely mesh 1, mesh 2, mesh 3, and mesh 4. The accuracy difference between the finest mesh and any other mesh can be determined using the following formula:(13)$$E=|\frac{M_{2}-M_{1}}{M_{1}}|\times 100\%$$

In the provided formula, *M*_1_ represents the finest mesh while *M*_2_ represents any other mesh. The inlet velocity is specified as 0.5 m/s, and the corresponding results are presented in Table 2.

The results show that the inlet and outlet pressure drop obtained from the numerical calculation of the model with a grid number of 1.479 million is the base, when the grid number of grids is 0.379 million, the relative error of the results is 1.04%; when the number of grids is 1.079 million, the relative error of the results is 0.23%. The results are more accurate when the dense, denser, and very dense grid models are used for numerical calculations, but they are more time-consuming. Therefore, on the basis of the accuracy and economy of the numerical calculation, the model with a grid number of 0.652 million was chosen. For the other models, the same method as case 0 was employed for grid independence verification in this simulation. Four different grids were selected and evaluated based on the parameter E. The final grid numbers used were 0.763 million, 0.754 million, 0.784 million, and 0.772 million.

### 2.5. Validation for Numerical Model

To validate the accuracy of the numerical calculation method employed in this paper, the computed results of the inlet and outlet pressure drop, as well as the fluid temperature difference, in the smooth channel (referred to as case 0) are compared with the corresponding theoretical calculations [26,27].

The theoretical formula for the pressure drop across the inlet and outlet of a rectangular micro-channel under laminar flow is given by:(14)$$\Delta p=\frac{(P_{0})\mu u_{in}L}{2D_{h}^{2}}+K\frac{\rho u_{in}^{2}}{2}$$
where *P*_0_ is Poiseuille’s number and *K* is the correction factor. *P*_0_ and *K* are calculated as:(15)$$P_{0}=96[1-1.3553\alpha +1.9467\alpha^{2}-1.7012\alpha^{3}+0.9564\alpha^{4}-0.2537\alpha^{5}]$$
(16)$$K=0.6796+1.2197\alpha +3.3089\alpha^{2}-9.5921\alpha^{3}+8.9089\alpha^{4}-2.9969\alpha^{5}$$

In this paper, the aspect ratio (*α*) of the rectangular micro-channel is assumed to be 0.5.

The theoretical equation for the temperature difference between the fluid inlet and outlet of the micro-channel proposed by Garimella and Singhal [28] is as follows:(17)$$\Delta T=T_{out}-T_{in}=\frac{qA}{\rho_{f,in}A_{in}U_{in}C_{p,f,in}}$$

*A* and *A_in_* are the areas of the bottom and inlet cross-sections of the micro-channel, respectively. The results of the numerical calculation of the pressure drop and fluid temperature difference between the inlet and outlet of the rectangular micro-channel at different Reynolds numbers are compared with the theoretical calculation. The comparison of our simulation results with the theoretical results from Steinke and Kandlikar [27] and Garimella and Singhal [28] is illustrated in Figure 4. It is evident that the discrepancy between our simulation results and the theoretical results is less than 10% for both pressure drop and fluid temperature difference, indicating that our code can be utilized with increased confidence. 

## 3. Results and Discussion

### 3.1. Flow Distribution

In this research, 3D numerical simulations for various geometric parameters are carried out to obtain the thermal performance of the micro-channels with the variable geometry rectangular fin. There are five input velocities taken into account: 0.6, 0.8, 1.0, 1.2, and 1.4 m/s. An in-depth analysis is completed on the corresponding pumping power and Reynolds number ranges. The subsequent paragraphs investigate the effects of these parameters on temperature distribution, pressure drop, and thermal resistance, revealing their respective impacts.

Understanding the flow structure within the micro-channel is crucial for assessing the performance of the MCHS. Thus, it is imperative to investigate the flow structure within the channel with a rectangular fin. Figure 5 illustrates the flow profile in the micro-channel with the variable geometry rectangular fin. Streamlines for each case are depicted in the x-z planes at a Reynolds number of 508.

It can be seen from the figures that the obstructions cause significant effects on the flow of the MCHS fluid. When encountering obstructions, the boundary layer of the fluid inside the flow channel is continuously disrupted, enhancing the mixing effect between the cold and hot fluids. The rectangular fin, located in the micro-channels, provides a different velocity distribution for the laminar flow profile than those of straight, smooth MCHS. Thus, the boundary layer cannot expand fully, and it is much less thin than smooth MCHS.

The presence of obstacles in MCHS leads to a significant increase in the Nusselt number. This is primarily attributed to the obstacles causing an acceleration of the coolant velocity in the regions between them, resulting in higher velocities compared to a straight and smooth MCHS configuration. Further, it can also be seen that the recirculation zone shows up in the backward part of the obstruction as a result of the effect of the block.

When the main flows traverse the rectangular fin, they deflect upwards in the rib upstream and create a recirculation zone downstream of the fins. The presence of a thin thermal boundary layer in the MCHS configuration can improve the mixing capability of both cold and hot fluids [28].

Comparing cases 0, 1, 2, 3, and 4 reveals that the depth of recirculation correlated with the separation of the main channel at the fin tip. Figure 6 illustrates the velocity distribution in the x-y plane at a height of z = 0.25 mm. In the case of case 0, it is evident that the velocity distribution exhibits maximum values near the center of the channel, gradually decreasing towards the walls. This is because the velocity of the fluid in the channel is greatest at the center due to the Bernoulli equation (i.e., the pressure is the lowest at the center). Therefore, the fluid is flowing faster at the center of the channel, resulting in a higher maximum velocity than at the edges. In the micro-channel with offset fins, the fluid experiences higher maximum velocities as it moves towards the sidewall without ribs. The fins create friction between the fluid and the wall, and that friction causes the velocity of the fluid to decrease. Without the fins, there is less friction, so the fluid will be able to move faster. Figure 6 demonstrates that the velocity within the recirculation zone is comparatively lower.

The velocity in a recirculation zone is usually very low due to the presence of vortices, which cause a high level of fluid separation. This low velocity is a result of the fact that the vortices in the zone act as a barrier, slowing down the flow and preventing it from exiting the zone. As a result, the velocity of the flow within the recirculation zone is generally much lower than the velocity of the flow outside of the zone [29,30]. Moreover, the presence of flow obstructions in the MCHS disrupts and re-establishes the coolant boundary layer, leading to a non-uniform velocity distribution. Even though a recirculating stream is apparent from behind the obstruction, the total heat transfer and thermal properties are enhanced.

### 3.2. Thermal Performance

In all cases, the Nusselt number is a non-dimensional parameter, and it is used to rate the heat transfer in the convective mode. Figure 7 shows the variation of the Nusselt number for various configurations of fins with the Reynolds number. As we increase the Reynolds number, the Nusselt number increases, and so does heat transfer. With the increase in Reynolds, the heat exchange efficiency is also improved, and more heat can be taken away by the fluid working medium in the unit time of the heat source. From Figure 7, it can also be found that for the micro-channel heat sinks with rectangular fins, when the Reynolds number is relatively small, the Nusselt number increases quickly. In contrast, when the Reynolds number is rather large, the Nusselt number increases relatively slowly. Furthermore, different models present different heat transfer capabilities.

The heat exchange capacity of MCHS is very important, which will directly affect the stability and safety of the heat source. Hence, for all micro-channel heat sink models at *u_in_* = 1.2 m/s, the temperature distribution in the flow passage is depicted in Figure 8. The plot clearly demonstrates that the temperature in the smooth micro-channel is considerably higher compared to the newly proposed model, providing evidence for the enhanced heat transfer achieved by the rectangular fins. Significantly, the presence of rectangular fins results in a considerable decrease in the temperature differential between the central region and the side wall of the channel. The observed temperature distribution can be attributed to the effective mixing of hot water near the channel’s side walls and cold water near its center. Case 4 has the lowest temperature of all configurations, which means it can take away more heat from the heat source. In addition, with a smooth channel, adding a rectangular fin greatly reduces the occurrence of local hotspots.

### 3.3. Pressure Drop

Figure 9 shows the pressure distribution of different channel models, including the smooth micro-channel. The observations from Figure 9 clearly indicate that the smooth channels exhibit lower pressure drop compared to the other channel models. This confirms that pressure losses in smooth channels primarily occur due to wall friction. However, in the case of the rectangular fin channel, additional pressure loss is introduced due to the generation of eddies in the liquid flow as it traverses over the ridges. The presence of the ridges in a fin channel leads to an increase in the surface roughness. As the fluid flows over the ridges, the turbulent eddies created by the features cause an increase in the frictional losses in the channel. This loss of energy increases the pressure drop significantly, leading to a higher required pressure to maintain the same flow rate. The pressure drop in case 4 is the highest among all configurations, primarily due to the presence of rectangular fins in this channel. These fins introduce additional flow restrictions, leading to greater resistance to fluid flow. As a result, the rectangular fins significantly contribute to the increased pressure drop in case 4 compared to the other configurations.

Additionally, Figure 10 displays the correlation between the Darcy friction factor and Reynolds number for cases 1–4 and the smooth micro-channel. The Darcy friction factor is an important dimensionless quantity used to determine the friction inside a channel flowing with a liquid. It is the ratio of the pressure drop over the fluid per unit length of the channel to the square of the average velocity of the fluid. From Figure 10, it is evident that the friction factor for cases 1–4 is more significant than that of smooth micro-channels because of obstruction to flow caused by fins. At the same time, it can also be seen from the figure that when the Reynolds number is less than 450, the friction factor decreases significantly, and after the Reynolds number is greater than 450, the declining trend of the friction factor becomes gentle. It is likely due to a threshold of the friction factor relative to the Reynolds number. The characteristic of this threshold makes the Reynolds number larger than the less obvious effect of the friction factor beyond this threshold. Within the entire range of Reynolds number variation, the friction factor value of case 3 is the maximum. This is mainly due to the structure of this channel, with more obstacles to impede the fluid flow.

### 3.4. Overall Thermal Performance

Figure 11, Figure 12 and Figure 13 depict the variations of *Nu/Nu*_0_, *f/f*_0_, and *PEC* as the Reynolds numbers range from 190 to 838. These figures serve to further evaluate the heat transfer performance of different micro-channel flow structures and compare them with the smooth micro-channel heat sink. As evident from Figure 11, the micro-channel with obstacles exhibits superior heat transfer performance compared to the smooth channel. Notably, within the range of Reynolds numbers less than 450, the micro-channel structure with obstacles demonstrates a more pronounced advantage in terms of thermal efficiency. As depicted in Figure 12, the newly proposed MCHS demonstrates a noticeable increase in the friction factor compared to the smooth micro-channel. Specifically, the friction factor of the proposed micro-channel heat sinks is 1.93–4.57 times higher than that of the smooth heat sink design. From the graph, it can also be seen that as the Reynolds number increases, the growth rate of the friction factor first increases and then decreases. Hence, It is evident from Figure 11 and Figure 12 that in the laminar flow stage, the heat transfer advantage of micro-channel structures with obstacles is obvious. The presence of obstacles can reduce the energy input to the heat transfer surface, increase the temperature difference, and thus increase the heat exchange efficiency. In addition, the flow from the upper layer is also restricted by the presence of obstacles, which makes the flow in the micro-channel more uniform, which is conducive to improving heat exchange efficiency. In order to assess the overall combined performance of the model designed in this article, the *PEC* parameter, which considers the heat transfer enhancement (*Nu/Nu*_0_) and the friction factor (*f/f*_0_), is evaluated for all cases. From Figure 13, as the inlet Reynolds number rises, the overall thermal performance of the obstruction-filled channels increases. Case 4 has the highest *PEC*, while case 1 has the lowest. This means that increasing the number of obstacles in the flow passage can improve the overall heat transfer performance of the heat sink. It is also apparent that as Re rises, *PEC* initially rises significantly, reaches a maximum, and then falls off quickly. The significant pressure drop experienced by the newly proposed micro-channel heat sink has led to a diminishing of its ability to effectively enhance heat transfer at higher Reynolds number conditions.

## 4. Conclusions

In this study, an analysis is conducted on different shapes of rectangular fins and Reynolds numbers. The findings reveal that the micro-channel heat sink with rectangular fins demonstrated significantly higher Nusselt numbers and friction factors compared to the smooth heat sink. Specifically, the Nusselt numbers were 1.40–2.02 times higher, while the friction factors were 2.64–4.33 times higher. These remarkable improvements in performance led to performance evaluation criteria ranging from 1.23–1.95.Open interrupted MHCS has higher heat transfer performance than rectangular MCHS and is an effective way to improve the thermal performance of this type of MCHS.The periodic truncation of the fins makes the velocity boundary layer of the fluid always in an alternating state of destruction and reconstruction, and a transverse recirculation zone is generated at the tail of the rectangular fins. Under the interaction of this effect, the degree of fluid disturbance in the open intermittent MCHS is intensified, and the heat transfer performance is greatly improved.All rectangular fin configurations had a higher friction factor since adding fins made the flow more restricted. Due to the biggest pressure drop, case 3 has the highest friction factor across the whole Re range. An increase in pressure drop results in a higher demand for pumping power.Among the various case configurations, case 4 exhibits superior thermal performance. However, it is accompanied by a notable disadvantage of high-pressure drop, resulting in an increased requirement for pumping power. To quantitatively evaluate the overall performance of the MCHS, we introduced a thermal enhancement factor. It is observed that case 3, due to its significant pressure penalty, has the lowest thermal enhancement factor across all Reynolds numbers compared to the other cases.For single-phase media, the use of rough surfaces (such as rectangular fins, grooves, interruptions, etc.), can enhance disturbances and mixing in the fluid. Secondly, the use of flow disturbance units can create secondary flow and enhance the mixing between the mainstream fluid and the boundary layer fluid, thus achieving the purpose of enhanced heat transfer. This is currently a very effective method for enhancing convective heat transfer.

## Figures and Tables

**Figure 1 micromachines-14-01818-f001:**
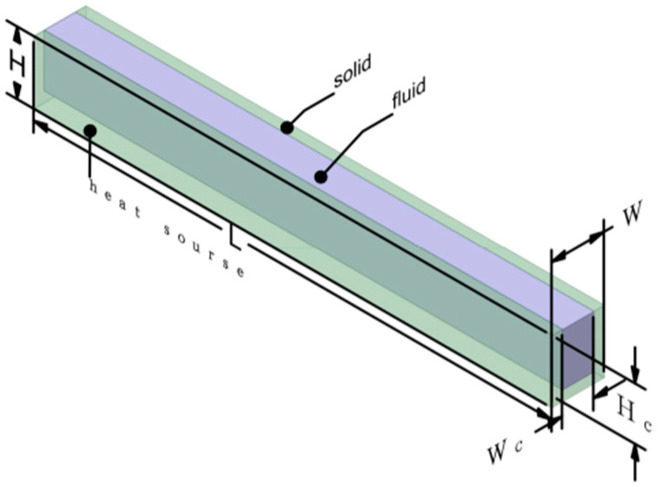
Schematic diagram of case 0.

**Figure 2 micromachines-14-01818-f002:**
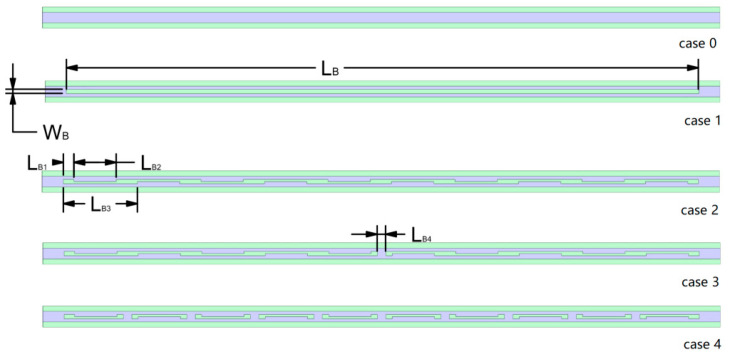
Top view of micro-channel heat sink: case 0–case 4.

**Figure 3 micromachines-14-01818-f003:**
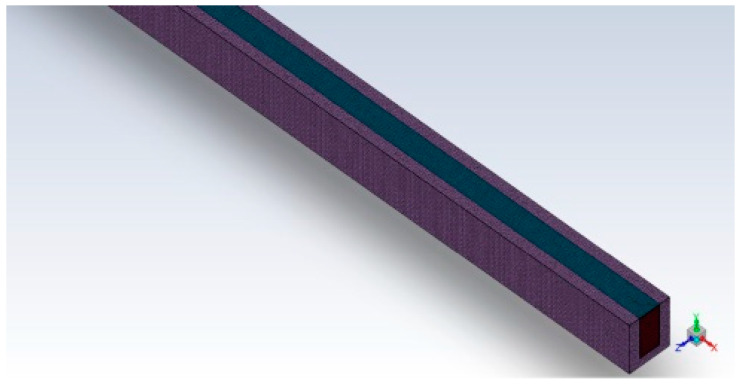
Computational meshes for case 0.

**Figure 4 micromachines-14-01818-f004:**
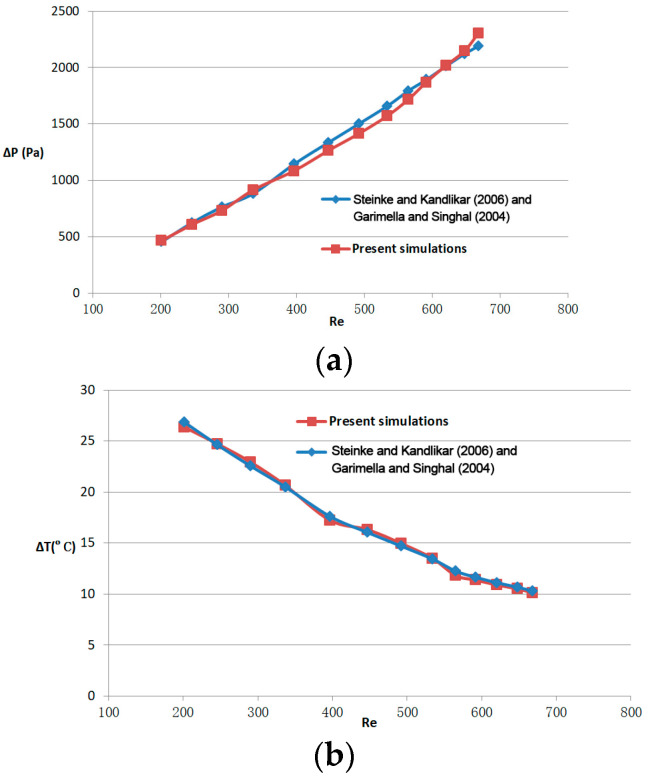
(**a**) The validation results for the pressure drop (ΔP) and (**b**) fluid temperature difference (ΔT) as a function of Reynolds number, based on case 0. Steinke and Kandlikar (2006) [27], Garimella and Singhal (2004) [28].

**Figure 5 micromachines-14-01818-f005:**
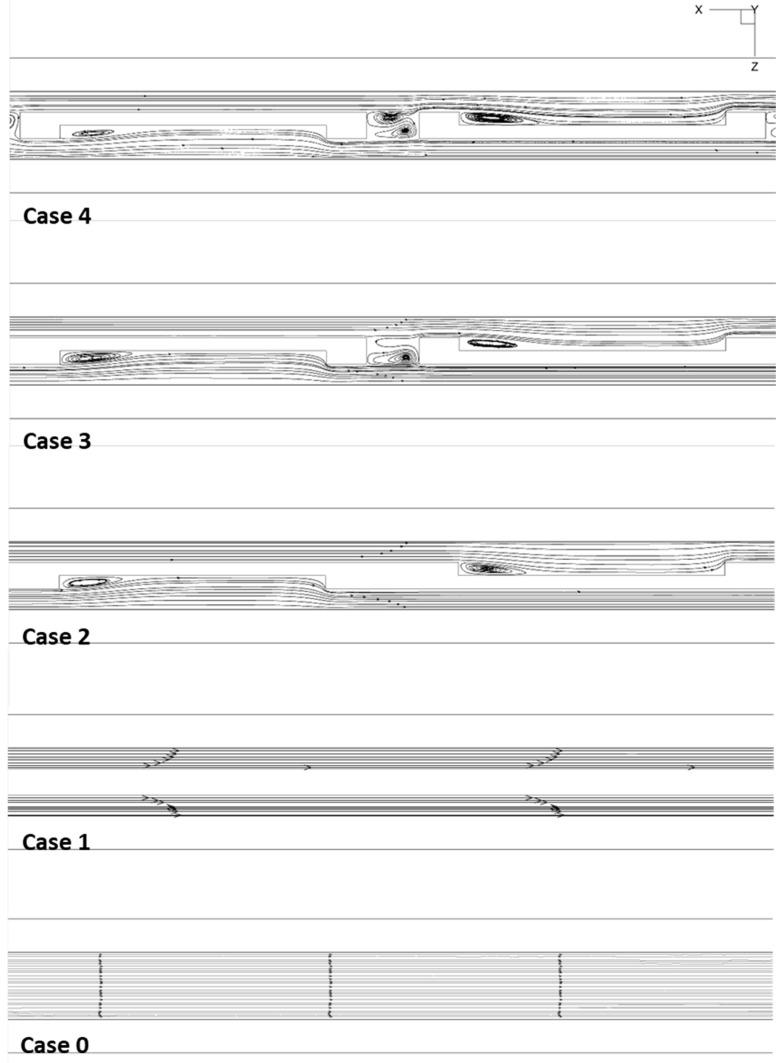
The distribution of streamlines in the x-z planes for all cases at a channel height of 0.25 mm. @*u_in_* = 1.2 m/s.

**Figure 6 micromachines-14-01818-f006:**
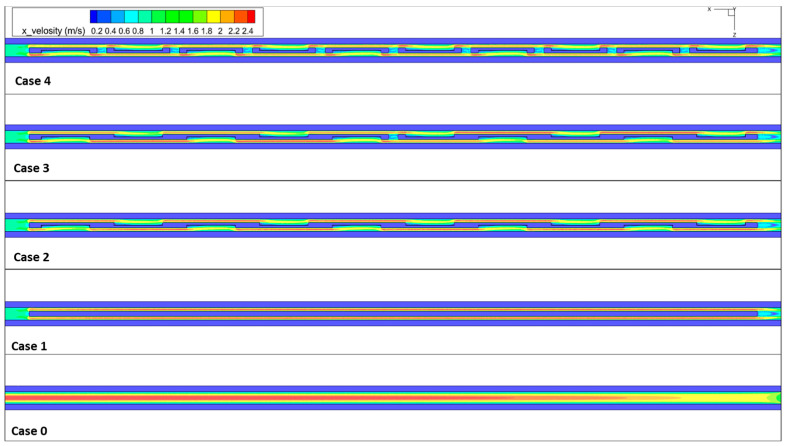
The velocity distribution in the x-z middle cross-section. @*u_in_* = 1.2 m/s.

**Figure 7 micromachines-14-01818-f007:**
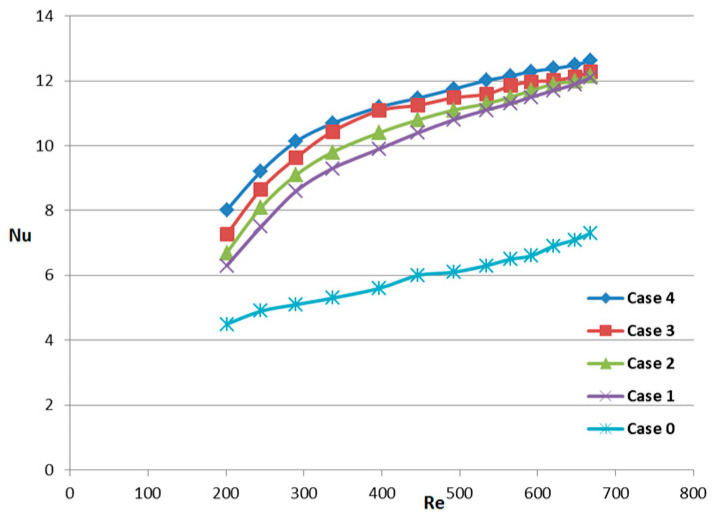
The relationship between the average Nusselt number for heat transfer in MCHS and the Reynolds number for all cases.

**Figure 8 micromachines-14-01818-f008:**
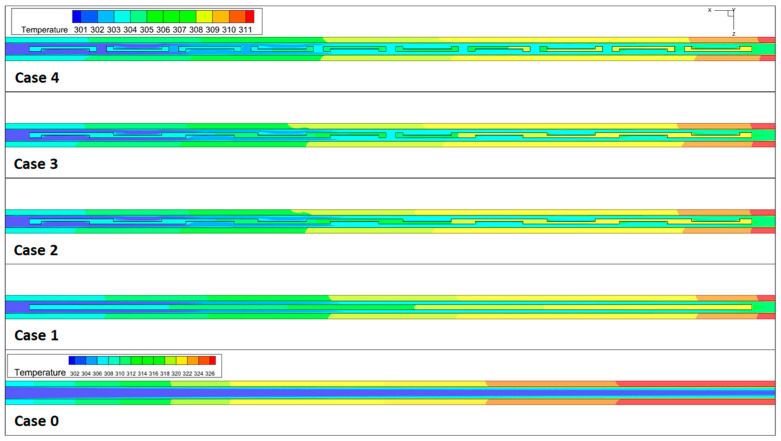
The temperature distribution of the x-z middle cross-section. @*u_in_* = 1.2 m/s.

**Figure 9 micromachines-14-01818-f009:**
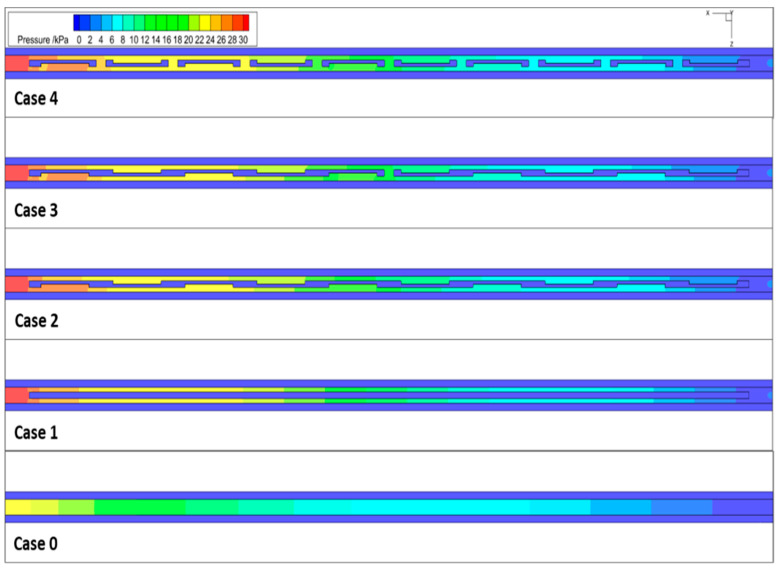
The pressure distribution of the x-z middle cross-section. @*u_in_* = 1.2 m/s.

**Figure 10 micromachines-14-01818-f010:**
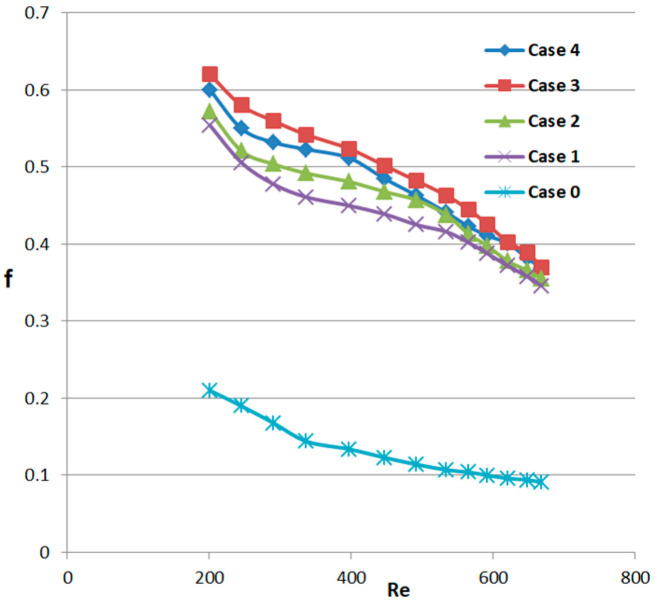
The relationship between the Darcy friction factors of micro-channel heat sinks and the Reynolds number for all cases.

**Figure 11 micromachines-14-01818-f011:**
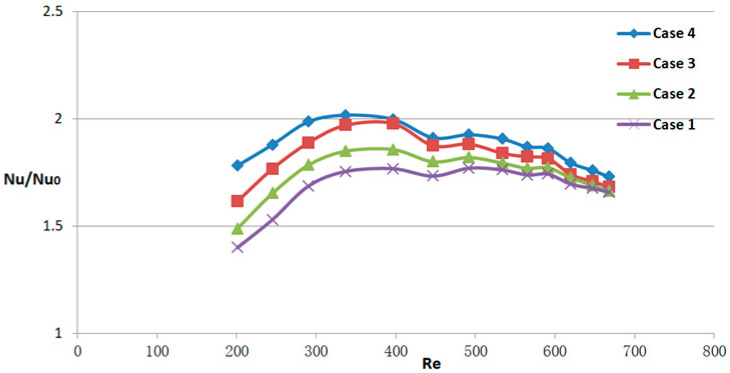
Heat transfer characteristics for MCHS with different design types, cases 1–4, based on case 0. The subscript 0 denotes case 0.

**Figure 12 micromachines-14-01818-f012:**
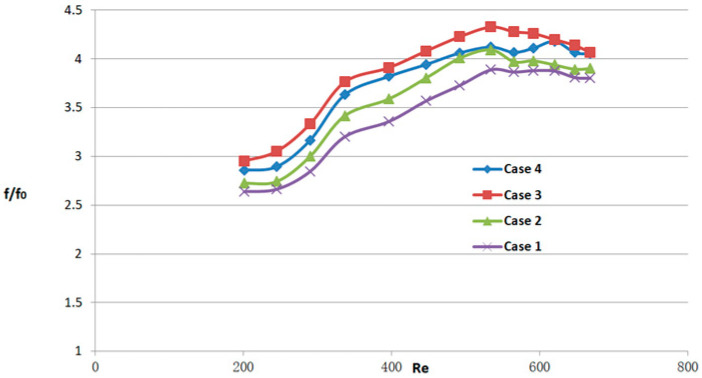
The Darcy friction factors for MCHS with flow obstacles, cases 1–4, based on case 0. The subscript 0 denotes case 0.

**Figure 13 micromachines-14-01818-f013:**
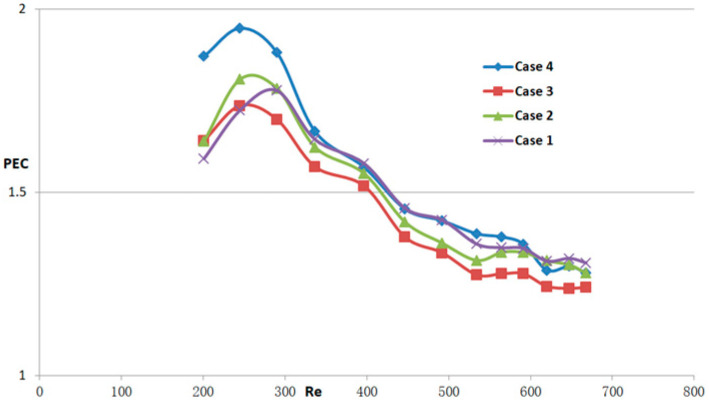
The PEC for MCHS with different design types, cases 1–4, based on case 0. The subscript 0 denotes case 0.

**Table 1 micromachines-14-01818-t001:** Dimensions of the MCHS (unit: mm).

H	W	L	L_B_	L_B1_	L_B2_	L_B3_	L_B4_	W_B_	W_C_	H_C_
1.2	1	32	5	5	20	35	4	0.25	0.65	1

**Table 2 micromachines-14-01818-t002:** Grid independence test, case 0, *u_in_* = 0.5 m/s.

Serial Number	Pressure Drop (Pa)	E
Mesh 1 (0.379 million grids)	859	1.04
Mesh 2 (0.652 million grids)	863	0.57
Mesh 3 (1.079 million grids)	866	0.23
Mesh 4 (1.479 million grids)	868	Baseline

## Data Availability

Not applicable.

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
