# Peer review of "Investigation of Hydrothermal Performance in Micro-Channel Heat Sink with Periodic Rectangular Fins"

_micromachines, 2023, doi:10.3390/mi14101818_

Round 1
Reviewer 1 Report (Previous Reviewer 2)
This paper conducts a evaluation of several micro-channel heat sink designs through numerical simulations, investigating various fin configurations. The flow characteristics, heat transfer and hydrodynamic efficiency are analysed. The findings are well presented. However, the discussions seem to revolve around specific design cases without providing overarching design guidelines for optimizing performance. Therefore I would recommend the authors address the following points:
1. When describing different cases (cases 1 to 4), it would be beneficial to emphasize the differences among different design and highlight the unique design features characterizing each design. For example, elucidate the differing attributes such as shape, length, or gap distance.
2. In conclusions, a more in-depth discussion of how these features affect hydrodynamics and thermal performance is neede. This would provide valuable high-level guidance regarding how alterations in design can affect the overall performance.
Author Response
- When describing different cases (cases 1 to 4), it would be beneficial to emphasize the differences among different design and highlight the unique design features characterizing each design. For example, elucidate the differing attributes such as shape, length, or gap distance.
Based on the reviewer's comments, the relevant modifications have been added to Section 2.2 of the paper, and they have been highlighted. The specific additions are as follows.
Case 1 in Figure 2 involves adding a rectangular fin with a length of LB inside the channel. Case 2 builds upon Case1 by incorporating periodic grooves into the rectangular fin with a length of LB2 and a depth of 1/2WB. Case 3 optimizes the rectangular fin by introducing a break with a length of LB4. Case 4 further optimizes Case 3 by applying periodic breaks with a length of LB4. The design of these grooves and breaks in Case 1-Case 4 generates vortices in the opposite direction to the mainstream flow within the channel. These vortices effectively mix the hot and cold fluids, continuously disrupting the thermal boundary layer and enhancing heat transfer capability.
- In conclusions, a more in-depth discussion of how these features affect hydrodynamics and thermal performance is needed. This would provide valuable high-level guidance regarding how alterations in design can affect the overall performance.
The reviewer's comments are very accurate, and the relevant modifications have been added in the conclusions section. The specific additions are as follows.
For single-phase media, the use of rough surfaces (such as rectangular fin, grooves, interruptions, etc.) can enhance disturbances and mixing in the fluid. Secondly, the use of flow disturbance units can create secondary flow and enhance the mixing between the mainstream fluid and the boundary layer fluid, thus achieving the purpose of enhanced heat transfer. This is currently a very effective method for enhancing convective heat transfer.
Reviewer 2 Report (New Reviewer)
1. The Abstract section is too simple, more key and qualitative conclusions, the data results, should be added in this section.
2. This paper focused on simulation comparison of five different periodic rectangular fins, but the main difference of five periodic rectangular fins is not detailed enough, the figure 2 is too small and not clear enough. The comparison table is suggested.
3. Horizontal coordinates and units cannot be seen clearly in Figure 4, and the subtitle is needed for each figure.
4. Some pictures, such as Figure 5 and 6, are so poor quality. They can not provide the useful information for reader.
5. The conclusions are ordinary, some data-driven conclusions should be analyzed.
No
Author Response
- The Abstract section is too simple, more key and qualitative conclusions, the data results, should be added in this section.
Based on the reviewer's comments, the relevant modifications have been added to the abstract section. The specific additions are as follows.
Based on the analysis of various rectangular fin shapes and Reynolds numbers in this study, the micro-channel heat sink with rectangular fins exhibits Nusselt numbers and friction factors that are 1.40-2.02 and 2.64-4.33 times higher, respectively, compared to the smooth heat sink. This significant improvement in performance results in performance evaluation criteria ranging from 1.23-1.95.
- This paper focused on simulation comparison of five different periodic rectangular fins, but the main difference of five periodic rectangular fins is not detailed enough, the figure 2 is too small and not clear enough. The comparison table is suggested.
According to the reviewer's comments, we have already added relevant content in the third paragraph of section 2.2 to provide a more detailed description of different models and explain the differences between them. The specific additions are as follows.
Case 1 in Figure 2 involves adding a rectangular fin with a length of LB inside the channel. Case 2 builds upon Case1 by incorporating periodic grooves into the rectangular fin with a length of LB2 and a depth of 1/2WB. Case 3 optimizes the rectangular fin by introducing a break with a length of LB4. Case 4 further optimizes Case 3 by applying periodic breaks with a length of LB4. The design of these grooves and breaks in Case 1-Case 4 generates vortices in the opposite direction to the mainstream flow within the channel. These vortices effectively mix the hot and cold fluids, continuously disrupting the thermal boundary layer and enhancing heat transfer capability.
- Horizontal coordinates and units cannot be seen clearly in Figure 4, and the subtitle is needed for each figure.
The reviewer's comments are very accurate, and the subtitle have been added in the relevant positions in the paper.
- Some pictures, such as Figure 5 and 6, are so poor quality. They can not provide the useful information for reader.
Thank you for the reviewer's comments. We have enlarged Figures 5-6 to ensure that readers can see the details of the images clearly, and also enlarged Figures 8-9. We hope this will better help readers understand the content of the paper.
- The conclusions are ordinary, some data-driven conclusions should be analyzed.
Thank you for the reviewer's comments. We have added data-driven conclusions in the conclusion section of the paper as described below.
In this study, an analysis was conducted on different shapes of rectangular fins and Reynolds numbers. The findings revealed that the micro-channel heat sink with rectangular fins demonstrated significantly higher Nusselt numbers and friction factors compared to the smooth heat sink. Specifically, the Nusselt numbers were 1.40-2.02 times higher, while the friction factors were 2.64-4.33 times higher. These remarkable improvements in performance led to performance evaluation criteria ranging from 1.23-1.95.
Round 2
Reviewer 2 Report (New Reviewer)
All comments have been revised.
NO
This manuscript is a resubmission of an earlier submission. The following is a list of the peer review reports and author responses from that submission.
Round 1
Reviewer 1 Report
The manuscript is worthy of being published. I suggest a y-axis range starting from 1 for Figures 11 to 13, as values below 1 do not make sense, or?
The manuscript is worthy of being published. I suggest a y-axis range starting from 1 for Figures 11 to 13, as values below 1 do not make sense, or?
Reviewer 2 Report
This work presents a simulation-based study of the performance of a micro-channel heat sink with different configurations of fins. The paper is overall well written and the results are clearly presented. However, I have the following concerns and suggestions that the authors should address in the revision:
1. Referring to case 0-case 4 in the abstract and in the introduction (line 124) without any context or parameters does not make sense. I would suggest removing it.
2. On the other hand, when describing the setup, the descriptions of cases 0-4 regarding how and why these fins are configured as shown in figure 2 are lacking (line 177).
3. For the grid independence study, the authors consider the base case 0 with a straight channel. I would like the authors to discuss whether the grid resolution is enough for other cases with fins, especially whether it can sufficiently resolve the boundary layer around the fins.
4. Line 295, “fluid coolant separates and damages, and accordingly boundaries …” I am not sure what “damage” means here; is this the right word to use? And do you mean “boundary layer” or “boundary”?
5. Line 323: describing a laminar flow as “turbulent and chaotic” is inaccurate.
Reviewer 3 Report
In this research paper, a computational fluid dynamics analysis was performed to investigate the laminar flow and heat transfer characteristics of five different configurations of a variable-geometry rectangular fin. The study utilized a water-cooled smooth MCHS as the basis. The results indicate that micro-channel heat sink with variable-geometry rectangular fin has better heat dissipation capacity than straight type micro-channel heat sink, but at the same time, it has larger pressure loss. Further, it is found that at a relatively small Reynolds number, micro-channel heat sink with variable-geometry rectangular fin has advantages in terms of overall cooling performance.
This paper cannot be accepted. The main reason is that the topic is poorly and parametrically addressed and has no relevance for the state of the art.
Other comments:
- figures are little and not visible.
- indices from measure units are not written correctly.
- typos and bad English
- Figure 4 is not clear and has poor design. The same for Fig 7, 10-13.
- The validation is not explained in terms of similarities with Ref 27 and 28.
- Results discussion is incorrect and no in-depth analysis is present. Plus, no comparison with state of the art was noticed.
Conclusion is not revealing authors contributions.
Very poor